# Antidepressant prescribing inequalities in people with comorbid depression and type 2 diabetes: A UK primary care electronic health record study

**Yutung Ng, Joseph F. Hayes, Annie Jeffery** *

Division of Psychiatry, University College London, London, United Kingdom

* annie.jeffery@ucl.ac.uk

## Abstract

### Aims

To compare the likelihood of being prescribed an antidepressant in depressed individuals with and without type 2 diabetes.

### Methods

We performed a matched cohort study using primary care record data from the UK Clinical Practice Research Datalink. We used multivariable logistic regression to compare antidepressant prescribing during the first five years of starting oral antidiabetic medication to a comparison group without type 2 diabetes, matched based on GP practice, age and sex. We performed subgroup analyses stratified by sex, age and ethnicity.

### Results

People with type 2 diabetes and depression were 75% less likely to be prescribed an antidepressant compared to people with depression alone (odds ratio (OR) 0.25, 95% confidence interval (CI) 0.25 to 0.26). This difference was greater in males (OR 0.23, 95% CI, 0.22 to 0.24), people older than 56 years (OR 0.23, 95% CI, 0.22 to 0.24), or from a minoritised ethnic background (Asian OR 0.14, 95% CI 0.12–0.14; Black OR 0.12, 95% CI 0.09–0.14).

### Conclusions

There may be inequalities in access to antidepressant treatment for people with type 2 diabetes, particularly those who are male, older or from minoritised ethnic backgrounds.

**Data Availability Statement:** Data cannot be shared publicly because it is person-level routinely collected medical data subject to data governance restrictions. These restrictions mean that data can

only be "accessed by bona fide researchers for public health research which is funded by trustworthy organisations" (see https://www.cprd.com/data-access). Data are available subject to protocol approval via CPRD's Research Data Governance Process which includes screening of researchers and funders, and quality assurance of the research protocol. Researchers can apply to access the data here: https://www.cprd.com/data-access.

**Funding:** The author(s) received no specific funding for this work.

**Competing interests:** The authors have declared that no competing interests exist.

## Introduction

People with type 2 diabetes are approximately three times more likely to suffer from depression than people without type 2 diabetes [1]. When the conditions are comorbid, depression has been shown to be associated with poor glycaemic control [2], the development of serious diabetic complications [3] and premature mortality [4]. Thus, the successful treatment of depression is important to both physical and mental health.

Antidepressants are recommended in UK guidelines to treat moderate to severe depression [5]. For people who have physical comorbidities, guidelines are vague, and not condition-specific–advising the clinician beware of possible drug-drug or drug-disease interactions [6]. In people with type 2 diabetes, antidepressants have been found in a Cochrane review to both reduce depressive symptoms and improve glycaemic control [7]. However, antidepressants have common side effects that may be of particular concern for people with type 2 diabetes; such as increased weight, cardiac symptoms, nausea and drug-drug interactions [8]. There is also very limited evidence concerning the effects of antidepressants on long-term physical health in people with type 2 diabetes [9]. This can make for difficult prescribing decisions, which could result in inappropriate prescribing or, alternatively, inequalities in access to treatment.

A retrospective cohort study using primary care data in the UK found that people with type 2 diabetes were more likely to be prescribed an antidepressant (excluding those prescribed at dosages for neuropathic pain) than people without type 2 diabetes [10]. However, this study did not measure whether or not patients had depression. Therefore, the higher rates of antidepressant prescribing are likely to reflect the higher prevalence of depression in people with type 2 diabetes. There is no other research in the UK, to the best of our knowledge, that investigates the difference in antidepressant prescribing in individuals with depression, between people with and without type 2 diabetes.

There is evidence in the general population that people with certain demographic characteristics are more likely to be prescribed antidepressants (female [11, 12]; older age [12]; and White ethnicity [13]). These inequalities could be of particular relevance to people with type 2 diabetes, which is more likely to affect males [14] and people from certain minoritised ethnic backgrounds [15]. It could also be more problematic in older adults who are more likely to have additional comorbidities [16]. However, none of the abovementioned demographic factors have been examined relative to antidepressant prescribing in individuals with and without type 2 diabetes.

We aimed to address this gap in the evidence using UK primary care data to compare, in people with depression, the likelihood of being prescribed an antidepressant in people with comorbid type 2 diabetes, and people without type 2 diabetes. We hypothesized that people with comorbid depression and type 2 diabetes would be more likely to be prescribed antidepressants than those without type 2 diabetes, due to the increased need to successfully treat depression in this patient group.

Our secondary aim was to investigate the sociodemographic factors that could influence the relationship between type 2 diabetes and antidepressant prescribing–in particular, sex, age, and ethnicity. We hypothesised that the increased prevalence of antidepressant prescribing in people with type 2 diabetes, would be greatest for individuals who were female, older and of White ethnicity.

## Methods

### Study design

We used a matched cohort study design to investigate the likelihood of individuals with depression being prescribed an antidepressant, in those with type 2 diabetes, compared to those without type 2 diabetes.

## Data source

We used primary care data from the UK Clinical Practice Research Datalink (CPRD) [17]. This data set encompasses anonymous electronic health records (EHRs) of approximately 60 million patients (16 million current patients) across the UK. The CPRD combines two databases, CPRD Gold and CPRD Aurum, which contain records from different EHR software (Vision and EMIS, respectively). We combined both CPRD Gold and Aurum in our analysis. The CPRD has been shown to be representative of the UK population with regards to sex, age and ethnicity [18, 19].

Data for this project were accessed from 1 January to 1 September 2022. Access to the data was approved by the Independent Scientific Advisory Committee of CPRD (protocol no. 21_001648). All data sent to the CPRD is anonymised.

## Study population

All participants in our study were registered with a CPRD GP practice for at least one year between the years 2000 to 2018 and had a record of depression (diagnosis (such as major depression or dysthymia), symptoms (such as "low mood") or process of care–such as "depression monitoring letter sent") during the EHR follow-up.

## Exposure and comparison

The exposed group comprised individuals with depression who also had type 2 diabetes and had started oral antidiabetic medication. We identified people with type 2 diabetes as those starting oral antidiabetic medication for the first time, as this was a standardised point in time to start follow-up. We required a 6-month period after their date of registration at a CPRD primary care practice without any oral antidiabetic prescriptions, before the date of their first oral antidiabetic prescription to indicate incident prescribing. We also required our exposed group to have at least 2 diagnostic tests for type 2 diabetes above the threshold used to indicate type 2 diabetes, as some antidiabetic medications (namely metformin) may be used to treat other conditions. We excluded people who were likely to have type 1 diabetes, by excluding those with an insulin prescription less than 6 months before the first oral antidiabetic prescription. We excluded people with probable gestational diabetes only, by excluding women who only had oral antidiabetic medication prescribed during periods of pregnancy. We set the index date of the exposed group as the date of their first oral antidiabetic prescription.

We included in our comparison group individuals with depression, but no record related to diabetes or any antidiabetic medication prescribed. We matched up to four individuals without type 2 diabetes randomly to people with type 2 diabetes, based on whether they were registered at the same GP practice, age (rolling 5 years) and sex. We chose a matching ratio of 1 to 4 controls to optimise statistical power [20] without needing to exclude individuals who did not have 4 controls eligible for matching. We used a random matching without replacement method, whereby each control could only be included once in the analysis, as this is more intuitive for interpretation [20].

We set the index date of the comparison group as the date of their match's first oral antidiabetic prescription.

In both the exposed and comparison groups, we excluded individuals prescribed antidepressants prior to the index date.

Fig 1 illustrates each stage of patient inclusion/exclusion.

We followed up all individuals in the study for 5 years or until the point of censoring. We censored anyone during this time who died or de-registered from their GP practice. We

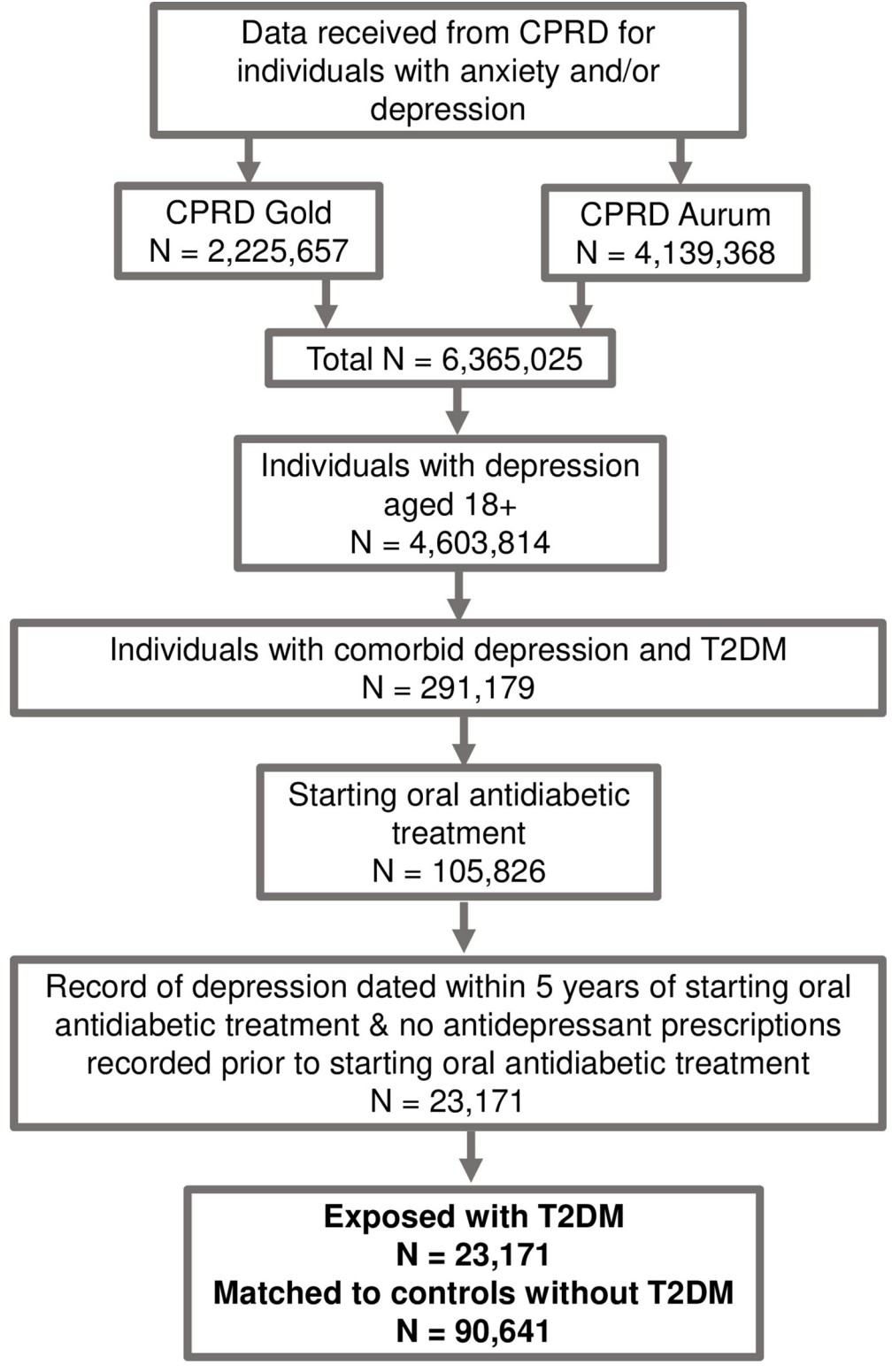

**Fig 1.**

required that both the exposed and comparison groups had a record of depression during their follow-up period.

## Outcome

Our outcome was the receipt of one or more antidepressant prescription(s) recorded during the follow-up period. We considered the following antidepressant medications: selective serotonin re-uptake inhibitors (citalopram, escitalopram, fluoxetine, fluvoxamine, paroxetine, sertraline), serotonin-noradrenaline reuptake inhibitors (duloxetine, venlafaxine), tricyclic antidepressants (amitriptyline > = 50mg, amoxapine, clomipramine, dosulepin, imipramine, lofepramine, maprotiline, nortriptyline > = 50mg, trimipramine), monoamine oxidase inhibitors (isocarboxazid, moclobemide, phenelzine, tranylcypromine) or other antidepressant agents (agomelatine, mianserin, mirtazapine, nefazodone, oxitriptan, reboxetine, trazodone, tryptophan). We excluded antidepressants amitriptyline and nortriptyline at lower doses prescribed for neuropathic pain.

## Confounders

We considered the following variables as confounders: sex (male or female), age (in years), ethnicity (categorised as Asian, Black, Mixed, Other, White, and Missing), and GP practice-level socioeconomic deprivation (using the index of multiple deprivation quintile [21]). We adjusted for confounders that were used for matching due to our mixed-ratio approach, in order to obtain unbiased odds ratios [20].

## Statistical analysis

We used logistic regression to model the association between type 2 diabetes and being prescribed antidepressant medication within 5 years of the study index date. We first performed a univariable logistic regression model and then added all aforementioned confounders in a multivariable model. Clustering by GP practice was accounted for by including GP practice ID as a strata term in multilevel models. We also performed interaction tests between the main exposure of type 2 diabetes, and age, sex and ethnicity. If we found any evidence of interactions (p-value <0.05), we performed subgroup analyses, stratifying by the aforementioned categories (for sex and ethnicity) and above/below the median age. All analyses were performed using R version 4.1.3. We used base R and the tidyverse package for all data management processes and analysis. We used the glm() function for logistic regression, with model parameter: family = binomial(link = 'logit').

## Ethical approval

The authors assert that all procedures contributing to this work comply with the ethical standards of the relevant national and institutional committees on human experimentation and with the Helsinki Declaration of 1975, as revised in 2008. All procedures involving human subjects/patients were approved by the Independent Scientific Advisory Committee of CPRD (protocol no. 21_001648). All data sent to the CPRD is anonymised and therefore consent is not required.

## Data availability

Data are available from the CPRD following study-specific protocol approval via CPRD's Research Data Governance Process.

**Table 1. Main baseline characteristics of the study population.**

|  | Total sample | Type 2 diabetes | Comparison without type 2 diabetes |
|---|---|---|---|
| **Total** | 113,812 | 23,171 | 90,641 |
| **Sex** | | | |
| Male | 58,688 (51.6%) | 11,998 (51.8) | 46,690 (51.2) |
| Female | 55,124 (48.4%) | 11,173 (48.2) | 43,951 (48.5) |
| **Age** | | | |
| Median age (Q1–Q3) | 56 (46–66 y) | - | - |
| > 55 y | 57,951 (50.9%) | 12,143 (52.4) | 45,808 (50.5) |
| < 56 y | 55,861 (49.1%) | 11,028 (47.6) | 44,833 (49.5) |
| **Ethnicity** | | | |
| White | 60,391 (53.1%) | 11,203 (48.3) | 49,188 (54.3) |
| Asian | 6,248 (5.5%) | 2,008 (8.7) | 4,240 (4.6) |
| Black | 2,539 (2.2%) | 775 (3.3) | 1,764 (1.9) |
| Mixed ethnicity | 615 (0.5%) | 132 (0.6) | 483 (0.5) |
| Others | 639 (0.6%) | 160 (0.7) | 479 (0.5) |
| Missing | 43,380 (38.1%) | 8,893 (38.4) | 34,487 (38.0) |

## Results

We included a total of 113,812 individuals with depression from 501 GP practices in the UK. Of these, 23,171 (20.3%) had type 2 diabetes and 90,641 (79.6%) did not have type 2 diabetes. Table 1 provides the details of the baseline characteristics of the patients included the analysis, stratified by whether or not they had type 2 diabetes. Overall, 48.4% of patients included were female, the median age was 56 years (interquartile range, 46–66), and the majority (53.1%) were of White ethnicity. Patients with type 2 diabetes were more likely to be of Asian (8.7% vs 4.6%) or Black (3.3% vs 1.9%) ethnicity.

The results for our comparison of antidepressant prescribing according to whether or not an individual had type 2 diabetes are presented in Table 2. Only 7.2% of people with type 2 diabetes were prescribed antidepressants, compared to 53.8% of people without type 2 diabetes. After adjusting for confounders, individuals with comorbid depression and type 2 diabetes were 75% less likely to be prescribed an antidepressant than those who had depression without type 2 diabetes (odds ratio (OR) 0.25, 95% CI, 0.25 to 0.26). There was no evidence of a change between the univariable and multivariable models.

We found evidence of an interaction between sex and type 2 diabetes, with regards to being prescribed an antidepressant ($p < 0.0001$). Therefore, we stratified our multivariable analysis by sex (Table 3). Being male further decreased the likelihood of being prescribed an antidepressant if the individual had type 2 diabetes, compared to if they did not have type 2 diabetes (77% less likely in men–OR 0.23, 95% CI, 0.22 to 0.24; compared to 72% less likely in women–OR 0.28, 95% CI, 0.27 to 0.29).

We found an interaction between age and type 2 diabetes, regarding antidepressants prescription ($p < 0.0001$). Therefore, we stratified our primary multivariable analysis by age

**Table 2. Results for the association between type 2 diabetes and antidepressant prescribing.**

|  | Prescribed antidepressants n (%) | Univariable analysis | Multivariable analysis |
|---|---|---|---|
|  |  | Odds ratio (95% CI) | Odds ratio (95% CI) |
| **Comparison group without type 2 diabetes** | 61,390 (67.7) | Reference | |
| **Exposed group with type 2 diabetes** | 8,170 (35.3) | 0.26 (0.25 to 0.27) | 0.25 (0.25 to 0.26) |

**Table 3. Subgroup analyses stratified by sex, age, and ethnicity.**

| | Comparison group without type 2 diabetes | Exposed group with type 2 diabetes |
|---|---|---|
| | Odds ratio (95% CI) | Odds ratio (95% CI) |
| **Sex** | | |
| Male | Reference | 0.23 (0.22 to 0.24) |
| Female | | 0.28 (0.27 to 0.29) |
| **Age** | Reference | |
| Over 55 years old | | 0.23 (0.22 to 0.24) |
| Less than 56 years old | | 0.28 (0.27 to 0.29) |
| **Ethnicity** | Reference | |
| White | | 0.30 (0.29 to 0.32) |
| Asian | | 0.14 (0.12 to 0.16) |
| Black | | 0.12 (0.09 to 0.14) |
| Mixed | | 0.15 (0.09 to 0.24) |
| Other | | 0.15 (0.10 to 0.22) |
| Missing | | 0.24 (0.23 to 0.25) |

(Table 3). Being over the age of 56 years further decreased the likelihood of being prescribed an antidepressant if the individual type 2 diabetes, compared to if they did not have type 2 diabetes (77% less likely in people aged over 56 years–OR 0.23, 95% CI, 0.22 to 0.24; compared to 72% less likely in people aged under 56 years–OR 0.28, 95% CI, 0.27 to 0.29).

We found an interaction between ethnicity and type 2 diabetes, with regards to being prescribed an antidepressant (p < 0.0001). Therefore, we stratified our primary multivariable analysis by ethnicity (Table 3). Being from an ethnic minority, compared to being of White ethnicity, further decreased the likelihood of being prescribed an antidepressant if the individual had type 2 diabetes, compared to if they did not have type 2 diabetes (70% lower in the majority of people of White ethnicity–OR 0.30, 95% CI, 0.29 to 0.32; compared to 86% lower in people of Asian ethnicity–OR 0.14, 95% CI, 0.12 to 0.16; 85% lower in people of Black ethnicity–OR 0.15, 95% CI, 0.09 to 0.24; 85% lower in people of Mixed ethnicity–OR 0.15, 95% CI 0.09 to 0.24; 85% lower in people of Other ethnicity–OR 0.15, 95% CI 0.10 to 0.22; and 76% lower in people with missing ethnicity).

## Discussion

### Summary

Our study was the first to investigate the difference in antidepressant prescribing in individuals with depression between those who did or did not have comorbid type 2 diabetes. We had hypothesised that people with type 2 diabetes would be more likely to be prescribed antidepressants, however, we found the opposite. People with comorbid depression and type 2 diabetes were 74% less likely to be prescribed antidepressants than people who had depression without type 2 diabetes, after adjusting for age, sex, ethnicity and GP practice. We found that the reduced likelihood of prescribing in people with type 2 diabetes was even further reduced in people who were male, older and from an ethnic minority background.

### Strengths and limitations

We used data from a large primary care database that is representative of the UK general population. This allowed us to investigate antidepressant prescribing as it happens in the real world, with a large enough sample size to investigate demographic interactions with type 2 diabetes

and antidepressant prescribing. Unlike other previous studies [10], all participants in our study had a record of depression. This means that it is not differences in depression rates driving our findings, but is truly difference in antidepressants prescribing for depression between individuals with type 2 diabetes and those without.

We included people with type 2 diabetes at the point where they start oral antidiabetic medication. Therefore our study is representative of individuals with type 2 diabetes, at the point where they start to require pharmaceutical intervention. It may not be generalisable to those who are managing their type 2 diabetes through lifestyle interventions such as diet and exercise, or to those with more complex type 2 diabetes.

We are confident in the validity of our antidepressant prescribing outcome, as antidepressant prescribing in the UK is through primary care, and all prescriptions are issued electronically and therefore automatically recorded on a patient's record. However, we were unable to see reasons for the decision to prescribe an antidepressant (or not), as these are not recorded in CPRD data. For example, we could not see whether a clinician considered a patient to be contraindicated for antidepressant treatment, whether antidepressant treatment was declined by the patient, or whether the option of antidepressant treatment was even discussed. Therefore, any interpretation as to why individuals with type 2 diabetes are prescribed less antidepressants than those without, can only be speculative. This is an important area for future research, including qualitative research on clinician and patient attitudes to antidepressant treatment in people with type 2 diabetes.

Antidepressant prescribing is only recommended for people with moderate to severe depression [5]. We were unable to adjust for depression severity, as this is not routinely recorded in CPRD data. We also included a wide range of clinical codes to identify people with depression. While this increases the sensitivity of our sample, we can consequently expect the sample to be heterogenous in terms of depression severity and subtype. This could mean that more severe depression in individuals without type 2 diabetes was driving the higher prevalence of antidepressant prescribing. Conceptually, it could be that people with type 2 diabetes were more likely to have mild depression recognized in primary care as they are already in contact with primary care through their type 2 diabetes; while people without type 2 diabetes might not access primary care unless they have more severe depression. However, we propose that this explanation is unlikely when there is evidence that people with type 2 diabetes in the UK often feel that their mental health is not acknowledged by clinicians [22] and when type 2 diabetes itself is associated with more severe depression [23].

We did not have patient-level data on sociodemographic characteristics. Although we accounted for deprivation at GP-practice level, there still will be socioeconomic variation between patients attending the same GP-practice. Therefore, there may have been unmeasured confounding by socioeconomic status. Socioeconomic deprivation is commonly associated with both antidepressant prescribing [24] and type 2 diabetes [25]. However, we would expect to see higher rates of antidepressant prescribing in people with type 2 diabetes, if socioeconomic status was causing considerable confounding. As such, the effect that we observed may have been underestimated.

## Explanation of findings and comparison to existing literature

There are a number of potential explanations as to why people with comorbid depression and type 2 diabetes, might be less likely to be prescribed antidepressants than people without type 2 diabetes. There are common side effects of antidepressants that may be particularly problematic for people with type 2 diabetes [11, 13]. Antidepressants are known to cause weight gain, which could exacerbate type 2 diabetes. Antidepressants are also known to cause cardiac

symptoms, which may be of concern to people with a condition such as type 2 diabetes, which can lead to cardiovascular disease. Although antidepressants were shown to improve glycaemic control in a systematic review of RCTs, they have also been shown in observational studies to be associated with hyperglycaemia [26] and starting insulin [27], as well as hypoglycaemia [28] (though all these may have been subject to confounding by indication). Despite the need to successfully treat depression in order effectively manage physical health in type 2 diabetes, without clear evidence on the benefits and harms of antidepressant medication in this patient group, and accompanying treatment guidelines, it is understandable that clinicians may be wary to prescribe. Indeed, there is evidence to suggest that clinicians under-prescribe in people who take multiple other medications, because of concern about the risk of drug interactions [29]. This explanation is further supported by our findings that older people with type 2 diabetes were even less likely to be prescribed antidepressants. Older individuals are more likely to both be prescribed multiple medications and experience medication side effects [30].

Conversely, people with type 2 diabetes are reported to experience a lack of acknowledgement of mental health concerns in clinical care [22]. While depression may be recorded on a patient's record, they might not have time to discuss antidepressant treatment or it might not be considered a priority for care. In addition, depression may be confused with diabetes-related distress [31], which is not recommended to be treated with antidepressant medication [32]. Alternatively, they could be more likely to receive psychological therapy instead of antidepressant treatment. However, this information is not available in primary care data. As a result of these, the likelihood of a clinician prescribing of antidepressants to people with type 2 diabetes who have depressive symptoms may be decreased. Our study found that the disparity in antidepressant prescribing in people with type 2 diabetes, was greatest in those who were male and from ethnic minorities. In the general population, both males and people from minoritised ethnic groups have been found to be less likely to receive prescriptions during their medical visits and to use antidepressants [12, 13]. Stigma about mental health could make people less likely to accept antidepressant medication [33]. This stigma is greater for males [34], older adults [35] and people from minoritised ethnic backgrounds [36]. Our study has shown that these inequalities are further increased, when the individual also has type 2 diabetes.

## Implications for research and practice

Our findings show that existing demographic inequalities related to antidepressant are exacerbated by having comorbid type 2 diabetes. Mental health needs to be prioritised in people with type 2 diabetes, to ensure that equal access to antidepressants is provided, when appropriate. Further research needs to explore the reasons for this considerable disparity in prescribing. Further research is also urgently required to clarify the benefits and harms of antidepressant prescribing in this patient group. This includes understanding for whom and when antidepressants might be appropriate or inappropriate.

## Acknowledgments

Designed the study (AJ, YN), curated the data (AJ), analysed the data (YN), interpreted the findings (all authors), drafted the manuscript (YN), edited the manuscript (all authors).

## Author Contributions

**Conceptualization:** Annie Jeffery.

**Data curation:** Annie Jeffery.

**Formal analysis:** Yutung Ng.

**Investigation:** Yutung Ng.

**Methodology:** Joseph F. Hayes, Annie Jeffery.

**Supervision:** Joseph F. Hayes, Annie Jeffery.

**Writing – original draft:** Yutung Ng.

**Writing – review & editing:** Yutung Ng, Joseph F. Hayes, Annie Jeffery.

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
