## [Decision Letter · Decision Letter 0]

14 May 2024

PONE-D-24-10322Antidepressant prescribing inequalities in people with comorbid depression and type 2 diabetes: a UK primary care electronic health record studyPLOS ONE

Dear Dr. Jeffery,

Thank you for submitting your manuscript to PLOS ONE. After careful consideration, we feel that it has merit but does not fully meet PLOS ONE’s publication criteria as it currently stands. Therefore, we invite you to submit a revised version of the manuscript that addresses the points raised during the review process.

 Both reviewers have highlighted that key information is absent from the paper and that this is required.

We look forward to receiving your revised manuscript.

Kind regards,

Kathleen Bennett

Academic Editor

PLOS ONE

https://journals.plos.org/plosone/s/file?id=ba62/PLOSOne_formatting_sample_title_authors_affiliations.pdf"

2.For studies involving third-party data, we encourage authors to share any data specific to their analyses that they can legally distribute. PLOS recognizes, however, that authors may be using third-party data they do not have the rights to share. When third-party data cannot be publicly shared, authors must provide all information necessary for interested researchers to apply to gain access to the data. (https://journals.plos.org/plosone/s/data-availability#loc-acceptable-data-access-restrictions) 

a) A description of the data set and the third-party source

b) If applicable, verification of permission to use the data set

c) Confirmation of whether the authors received any special privileges in accessing the data that other researchers would not have

d) All necessary contact information others would need to apply to gain access to the data

3.Your ethics statement should only appear in the Methods section of your manuscript. If your ethics statement is written in any section besides the Methods, please move it to the Methods section and delete it from any other section. Please ensure that your ethics statement is included in your manuscript, as the ethics statement entered into the online submission form will not be published alongside your manuscript. 

Reviewers' comments:

Reviewer's Responses to Questions

**Comments to the Author**

1. Is the manuscript technically sound, and do the data support the conclusions?

Reviewer #1: Yes

Reviewer #2: Partly

2. Has the statistical analysis been performed appropriately and rigorously? 

Reviewer #1: Yes

Reviewer #2: No

3. Have the authors made all data underlying the findings in their manuscript fully available?

Reviewer #1: Yes

Reviewer #2: No

4. Is the manuscript presented in an intelligible fashion and written in standard English?

Reviewer #1: Yes

Reviewer #2: Yes

5. Review Comments to the Author

Reviewer #1: Please read my comments in the attached PDF.

Needs further clarification about those who are diagnosed with T2DM but not prescribed anti-diabetic medications

Statistical software used needs to be added

How would you explain those who were prescribed antidepressant for painful neuropathy rather than depression?

Add the limitations of the CPRD data - multiple codes are used in primary care electronic health record for depression which may have confounded the results.

Reviewer #2: This paper addresses an important topic of antidepressant prescribing inequalities using a large primary care dataset.

It is generally well-written, however some key information is missing to assess the findings of this paper e.g. how many people were removed before analysis because of death or de-registering from their GP? It is mentioned that these were censored in which case why not include them up to the point of censoring in a Cox regression? Why not allow for varying lengths of follow up? What is the justification for using a logistic regression with a fixed follow up of 5 years? A flow chart would be useful showing how many people are excluded at each stage from the definition of the study population to the final dataset for analysis. This is important to assess the sentence that the database is representative of the UK general population.

More detail is required on the method used for matching. What is the justification for using 4:1 matching?

The statistical analysis section is brief. What statistical software was used? What procedures within the software?

Age, sex and general practice were used for matching and also used as confounders in the analysis. This approach should be justified and referenced e.g.

Sjölander, A., & Greenland, S. (2013). Ignoring the matching variables in cohort studies–when is it valid and why?. Statistics in medicine, 32(27), 4696-4708.

What is the rate of prescribing of antidepressants in each group over the 5-year follow up? Results are reported that people with type 2 diabetes are less likely to be prescribed but the overall rates for each group would be useful.

How many general practices were included in the final dataset? The hierarchical structure of the data with patients within practices was accounted for by using a multilevel model but no results are reported on this level.

In Table 3 the reference group is the group with type 2 diabetes?

6. PLOS authors have the option to publish the peer review history of their article (what does this mean?). If published, this will include your full peer review and any attached files.

Reviewer #1: **Yes: **Debasish Kar

Reviewer #2: No

---

## [Author Response · Author response to Decision Letter 0]

5 Jul 2024

Response to reviewers

We would like to thank the reviewers for their helpful comments. We have revised the manuscript accordingly. Please see response to the individual reviewer comments, with amendment page/line numbers corresponding to the document with tracked changes. Please see attached version of response to reviewers - this is formatted in a way that is easier to follow. 

Reviewer 1, comments on document:

1. Introduction page 3, line 9: “be ware” changed to “be aware”

2. Introduction page 3, lines 18-19: Thank you for pointing out that some antidepressants are prescribed to treat neuropathic pain. Pal et al excluded amitriptyline and nortriptyline prescribed at the lower doses expected for this indication. We have added this information (addition in italics): “that people with type 2 diabetes were more likely to be prescribed an antidepressant (excluding those prescribed at dosages for neuropathic pain) than people without type 2 diabetes.

3. Methods, Exposure and comparison, page 4, lines 26-27: In response to your query about people who had type 2 diabetes managed by lifestyle interventions. We have added to the first line describing the exposed group that we were looking at those who had started oral antidiabetic medication (addition in italics): “The exposed group comprised individuals with depression who also had type 2 diabetes and had started oral antidiabetic medication.” We have also noted this as a limitation to the generalisability of the study in our limitations section, page 8, lines 10-14: “We included people with type 2 diabetes at the point where they start oral antidiabetic medication. Therefore our study is representative of individuals with type 2 diabetes, at the point where they start to require pharmaceutical intervention. It may not be generalisable to those who are managing their type 2 diabetes through lifestyle interventions such as diet and exercise, or to those with more complex type 2 diabetes.”

4. Methods, Confounders, pages 5-6, lines 33-1: Many apologies for a mistake in the text and limitations section. We did not have patient-level information on deprivation. We previously had in our confounders section written that we adjusted for GP practice as a confounder. We have corrected this to say GP-level deprivation (addition in italics): “GP practice-level socioeconomic deprivation (using the index of multiple deprivation quintile)”. We also used a multilevel model to account for clustering at GP-level, which is described in our Statistical analysis section, page 6, lines 7-8. We have updated Table 1 to add the breakdown of deprivation quintile by exposure group. We have also corrected this in our Discussion, Strengths and limitations section on page 9, lines 4-6. Here we previously said: “Socioeconomic characteristics are also not recorded in CPRD data. Although we matched participants based on GP practice, which would have accounted for socioeconomic characteristics to an extent, there could still be a considerable amount of unmeasured confounding from socioeconomic status.” We now say this, page 9, lines 6-9: “We did not have patient-level data on sociodemographic characteristics. Although we accounted for deprivation at GP-practice level, there still will be socioeconomic variation between patients attending the same GP-practice. Therefore, there may have been unmeasured confounding by socioeconomic status.”

5. Methods, statistical analysis, page 6, line 6: All analysis used logistic regression. We have replaced the term: “univariable analysis” with “univariable logistic regression model”.

6. Methods statistical analysis, page 6, line 11: We have added the statistical software used: “All analyses were performed using R version 4.1.3.”

7. Results, page 7, line 18: Thank you for noticing that we did not state the reference group, we have added this in the text: “compared to being of White ethnicity”. 

8. Strengths and limitations: Thank you for pointing out that not all people in primary care have a diagnosis recorded. We have added further information on how people with depression were identified in the Methods section page 4, lines 22-23 (additions in italics): “a record of depression (diagnosis (such as major depression or dysthymia), symptoms (such as “low mood”) or process of care – such as “depression monitoring letter sent”). We have added an additional sentence to our discussion of potential confounding by depression severity, highlighting the strengths and limitations of this inclusion criteria, Discussion, Strengths and limitations, page 8, lines 26-29: We also included a wide range of clinical codes to identify people with depression. While this increases the sensitivity of our sample, we can consequently expect the sample to be heterogenous in terms of depression severity and subtype.”

Further comments from Reviewer 1:

1. “Needs further clarification about those who are diagnosed with T2DM but not prescribed anti-diabetic medications” – please see response item 3 above from comments on document 

2. “Statistical software used needs to be added” – please see response item 6 above from comments on document

3. How would you explain those who were prescribed antidepressant for painful neuropathy rather than depression? Thank you for pointing this out – we have added a sentence explaining our approach to this in our Methods, Outcome, page 5, lines 29-30: “We excluded antidepressants amitriptyline and nortriptyline at lower doses prescribed for neuropathic pain.”

4. “Add the limitations of the CPRD data - multiple codes are used in primary care electronic health record for depression which may have confounded the results.” – Please see response item 8 above from comments on document. We have now explicitly stated the limitation from the diverse list of depression codes included. This now provides context for the discussion that we already had on confounding by depression severity in our Methods, Strengths and limitations section, pages 8-9, lines 29-2: “This could mean that more severe depression in individuals without type 2 diabetes was driving the higher prevalence of antidepressant prescribing. Conceptually, it could be that people with type 2 diabetes were more likely to have mild depression recognized in primary care as they are already in contact with primary care through their type 2 diabetes; while people without type 2 diabetes might not access primary care unless they have more severe depression. However, we propose that this explanation is unlikely when there is evidence that people with type 2 diabetes in the UK often feel that their mental health is not acknowledged by clinicians (20) and when type 2 diabetes itself is associated with more severe depression (21).”

Reviewer 2 comments:

“This paper addresses an important topic of antidepressant prescribing inequalities using a large primary care dataset. It is generally well-written, however some key information is missing to assess the findings of this paper e.g.”…

1. “how many people were removed before analysis because of death or de-registering from their GP? It is mentioned that these were censored in which case why not include them up to the point of censoring in a Cox regression? Why not allow for varying lengths of follow up? What is the justification for using a logistic regression with a fixed follow up of 5 years? A flow chart would be useful showing how many people are excluded at each stage from the definition of the study population to the final dataset for analysis. This is important to assess the sentence that the database is representative of the UK general population.”

Apologies for the confusion here. When we stated that people were censored because of death or de-registration, we did not mean that they were excluded from the study. We included people up to the point of censoring, allowing for varying lengths of follow-up, as you recommend here. We have amended the text in our Methods, Exposure and comparison, page 5, lines 18-20, with the following additions to make this more explicit (additions in italics): “We followed up all individuals in the study for 5 years or until the point of censoring. We censored anyone during this time who died or de-registered from their GP practice. We required that both the exposed and comparison groups had a record of depression during their follow-up period.” We have added a flowchart showing the points at which people were excluded from the study (Fig 1). And noted this in our methods, page 5, line 17: “Fig 1 illustrates each stage of patient inclusion/exclusion.”

2. “More detail is required on the method used for matching. What is the justification for using 4:1 matching?”

We have added additional detail to our description of the matching, in our Methods section page 5, lines 9-12 (added detail in italics): “We included in our comparison group individuals with depression, but no record related to diabetes or any antidiabetic medication prescribed. We matched up to four individuals without type 2 diabetes randomly to people with type 2 diabetes, based on whether they were registered at the same GP practice, age (rolling 5 years) and sex. We chose a matching ratio of 1 to 4 controls to optimise statistical power (20) without needing to exclude individuals who did not have 4 controls eligible for matching. We used a random matching without replacement method, whereby each control could only be included once in the analysis, as this is more intuitive for interpretation (20).

3. “The statistical analysis section is brief. What statistical software was used? What procedures within the software?”

We have added the following detail on page 6, lines 11-13: “All analyses were performed using R version 4.1.3. We used base R and the tidyverse package for all data management processes and analysis. We used the glm() function for logistic regression, with model parameter: family=binomial(link='logit').”

4. Age, sex and general practice were used for matching and also used as confounders in the analysis. This approach should be justified and referenced e.g. Sjölander, A., & Greenland, S. (2013). Ignoring the matching variables in cohort studies–when is it valid and why?. Statistics in medicine, 32(27), 4696-4708.

As we used a variable matching ratio (i.e. up to four, rather than four) adjusting for matching variables is necessary for an unbiased estimate of the odds ratio. We have added this detail to our text on confounders, with a corresponding reference, pages 5-6, lines 33-2: “We adjusted for confounders that were used for matching due to our mixed-ratio approach, in order to obtain unbiased odds ratios (20).”

5. What is the rate of prescribing of antidepressants in each group over the 5-year follow up? Results are reported that people with type 2 diabetes are less likely to be prescribed but the overall rates for each group would be useful.

Thank you for this suggestion. We have added this to Table 2, and included it in our Results narrative on pages 6-7, lines 32-1: “Only 7.2% of people with type 2 diabetes were prescribed antidepressants, compared to 53.8% of people without type 2 diabetes.”

6. How many general practices were included in the final dataset? The hierarchical structure of the data with patients within practices was accounted for by using a multilevel model but no results are reported on this level.

We have added this to our results narrative, page 6, line 25: “from 501 GP practices in the UK”

7. “In Table 3 the reference group is the group with type 2 diabetes?” 

Many apologies, and thank you for noticing this mistake. This table has been corrected – the reference group is people without type 2 diabetes.

Further changes: The following have been added to the Methods section, page 6, lines 14-22:

“Ethical approval

The authors assert that all procedures contributing to this work comply with the ethical standards of the relevant national and institutional committees on human experimentation and with the Helsinki Declaration of 1975, as revised in 2008. All procedures involving human subjects/patients were approved by the Independent Scientific Advisory Committee of CPRD (protocol no. 21_001648). All data sent to the CPRD is anonymised and therefore consent is not required.

Data availability

Data are available from the CPRD following study-specific protocol approval via CPRD's Research Data Governance Process.”

---

## [Editor Report · Decision Letter 1]

10 Jul 2024

PONE-D-24-10322R1Antidepressant prescribing inequalities in people with comorbid depression and type 2 diabetes: a UK primary care electronic health record studyPLOS ONE

Dear Dr. Jeffery,

Thank you for submitting your manuscript to PLOS ONE. After careful consideration, we feel that it has merit but does not fully meet PLOS ONE’s publication criteria as it currently stands. Therefore, we invite you to submit a revised version of the manuscript that addresses the points raised during the review process.

 In particular, there is just one outstanding query in relation to the response and updated Table 2. Please check the figures provided in the table.

We look forward to receiving your revised manuscript.

Kind regards,

Kathleen Bennett

Academic Editor

PLOS ONE

Journal Requirements:

Additional Editor Comments:

The authors have addressed the reviewers comments appropriately. Just one outstanding query on the figures include in Table 2 for the prescribed antidepressants n(%) for the two groups. The number is 61,390 for the comparison group (which would equate to 61,390/90,641 = 67.7% (not 53.9%); similarly for the type 2 diabetes group the number on antidepressants is 8170 but the % would be 8170/23171 = 35.3%. Can the authors clarify and check this?

---

## [Author Response · Author response to Decision Letter 1]

23 Jul 2024

Thank you for noticing the error in Table 2. This is now updated with the correct percentages.

---

## [Editor Report · Decision Letter 2]

7 Aug 2024

Antidepressant prescribing inequalities in people with comorbid depression and type 2 diabetes: a UK primary care electronic health record study

PONE-D-24-10322R2

Dear Dr. Jeffery,

We’re pleased to inform you that your manuscript has been judged scientifically suitable for publication and will be formally accepted for publication once it meets all outstanding technical requirements.

Kind regards,

Eyob Alemayehu Gebreyohannes, PhD

Academic Editor

PLOS ONE

---

## [Editor Report · Acceptance letter]

14 Aug 2024

PONE-D-24-10322R2 

PLOS ONE

Dear Dr. Jeffery, 

I'm pleased to inform you that your manuscript has been deemed suitable for publication in PLOS ONE. Congratulations! Your manuscript is now being handed over to our production team.

Kind regards, 

on behalf of

Dr. Eyob Alemayehu Gebreyohannes 

Academic Editor

PLOS ONE